# Sociodemographic and Clinical Profile of Long COVID-19 Patients, and Its Correlation with Medical Leave: A Comprehensive Descriptive and Multicenter Study

**DOI:** 10.3390/healthcare11192632

**Published:** 2023-09-27

**Authors:** Esperanza Romero-Rodriguez, Luis Angel Perula-de Torres, Jaime Monserrat-Villatoro, Jesus Gonzalez-Lama, Ana Belen Carmona-Casado, Antonio Ranchal-Sanchez

**Affiliations:** 1Maimonides Institute for Biomedical Research of Córdoba (IMIBIC), Reina Sofía University Hospital, Córdoba University, 14004 Córdoba, Spain; espe_mrr@hotmail.com (E.R.-R.); luisangel.perula@gmail.com (L.A.P.-d.T.); jaime.monserrat.sspa@juntadeandalucia.es (J.M.-V.); anabelen.carmona@imibic.org (A.B.C.-C.); 2Córdoba and Guadalquivir Health District, 14011 Córdoba, Spain; 3“Matrona Antonia Mesa Fernández” Health Center, Cabra Clinical Management Unit, AGS South of Córdoba, 14940 Córdoba, Spain; 4Department of Nursing, Pharmacology and Physiotherapy, Faculty of Medicine and Nursing, University of Cordoba, 14004 Córdoba, Spain

**Keywords:** comorbidity, gender equity, post-acute COVID-19 syndrome, quality of life, sick leave

## Abstract

The persistent condition of COVID-19 is characterized by a wide range of symptoms that have had a significant impact on both the health status and occupational life of the population. In this observational and multicenter study, the relationship between the sociodemographic and clinical profile of Spanish patients diagnosed with long COVID, and the work-related disability resulting from this pathology was analyzed. The analysis included 689 responses. A descriptive analysis of the variables recorded was performed, together with a bivariate analysis to determine associations between work-related disability and variables such as gender, age, health status, disabling symptoms or comorbidities. The results obtained highlight fatigue and lack of concentration (brain fog) as the most incapacitating symptoms among patients diagnosed with long COVID. Multivariate analysis revealed that time since diagnosis (OR: 0.57, CI95%: 0.36–0.89, *p*: 0.013), concomitant renal insufficiency (OR: 4.04, CI95%: 1.42–11.4, *p*: 0.008), and symptoms like fatigue (OR: 0.56, CI95%: 0.33–0.99) and tremors (OR: 2.0, CI95%: 1.06–3.69, *p*: 0.029), were associated with work-related disability. These findings highlight the need to improve the health and work-related management of this condition in the healthcare system. Besides risk factor control, it is suggested to pay special attention to determining the appropriate timing of medical leave work reintegration, along with coordination between primary care and occupational health services to ensure the gradual and tailored return of patients with long COVID to the workforce.

## 1. Introduction

Throughout the SARS-CoV-2 pandemic, a considerable heterogeneity in the symptomatic presentation of affected patients was observed, with well-known symptoms including fever, loss of smell and taste, and gastrointestinal symptoms, among others [1]. These symptoms can vary in duration, categorizing as post-acute COVID-19 if persisting for more than 3 weeks, and as persistent COVID-19 if lasting over 12 weeks [2]. The term “long COVID” refers to symptoms following the viral infection that persist beyond the acute phase or reappear after prior infection [3]. Persistent COVID can occur in individuals with severe symptoms who were hospitalized during the acute phase, as well as those with milder symptoms, as its onset is not linked to the initial infection’s severity [4].

Given the lack of consensus in terminology, the WHO recommends referring to this condition as the “post-COVID-19 condition” [5]. Additionally, the National Institute for Health and Care Excellence (NICE) defines symptomatic phases as acute COVID-19 (for up to 4 weeks), ongoing symptomatic COVID-19 (from 4 to 12 weeks), and post-COVID-19 syndrome (signs and symptoms that develop during or after an infection consistent with COVID-19 that continue for more than 12 weeks and are not explained by an alternative diagnosis). In addition to the clinical case definitions, the term ‘long COVID’ is commonly used to describe signs and symptoms that continue or develop after acute COVID-19. It includes both ongoing symptomatic COVID-19 and post-COVID-19 syndrome [2,6]. In order to address this matter, we have reached a consensus regarding the terminology employed, thereby establishing the term post-acute sequelae of SARS-CoV-2 (PASC) for referring to long COVID.

Existing prevalence studies predominantly suggest that around 10% of SARS-CoV-2 infected patients experience PASC, with the prevalent profile being middle-aged women without prior health issues, though factors such as obesity, comorbidity, and hospitalization increase the likelihood of PASC [7,8,9]. However, this figure varies based on demographic factors. A Chinese study reveals a prevalence of 63% among 1733 hospitalized patients, while Bliddal et al. indicate a prevalence of 36% in 198 Danish non-hospitalized patients, with 44% being women and 24% men. Moreover, this article establishes a significant relationship between body mass index and gender with the likelihood of PASC [10,11]. In countries like Italy, even higher prevalence is reported, with 87.4% of 143 patients exhibiting persistent symptoms after two months of follow-up [4].

Approximately 200 different symptoms have been described in PASC, with the most frequently reported being fatigue, headache, impaired ability to concentrate, dyspnea, hair loss, and, occasionally, persistent pain without demonstrated organic impairment, such as myalgias, muscle pain, and generalized hyperalgesia [3,9].

In addition to physical symptoms, the emotional and social impact of the disease on these patients is noteworthy, potentially leading to incapacitation in their daily lives and subsequently affecting their quality of life [12]. Patients often express fear and concern about not regaining their previous health and life, facing limitations in their environment and social activities that they are unable to engage in due to physical or psychological distress [7,13]. This is affirmed by studies indicating that after 6 months from the acute phase, 85% of patients report their health status as moderate to poor [7].

Work-related incapacity is another concern for patients suffering from PASC inducing anxiety and stress due to the potential of job loss or early retirement, with personal, economic, and social repercussions for both the individual and society [7]. Various studies show that at least half of PASC patients experience temporary work incapacity or reduced usual work activities due to the inability to perform their previous tasks fully [14,15]. Furthermore, loss of or incapacity to perform work can lead to diminished quality of life, distorting self-concept and self-esteem, impeding social relationships, and ultimately diminishing both personal and social well-being [16,17].

In this context, few studies have investigated the occupational impact of long COVID patients. The objective of this study was to describe the sociodemographic and clinical profile of Spanish people suffering PASC and analyze its association with work-related incapacity.

## 2. Materials and Methods

### 2.1. Study Design

Descriptive observational multicenter study conducted in Spain.

### 2.2. Study Population

The study included the general adult population receiving care within the national health system. Inclusion criteria were as follows: (a) residency in Spain; (b) age 14 years or older; (c) laboratory-diagnosed acute COVID-19 infection; (d) meeting PASC (5); (e) providing consent to participate in the research study.

### 2.3. Sample Size

Given the nature of this survey, a sample size was not estimated; rather, the aim was to attain the highest number of subjects during the study’s recruitment period.

### 2.4. Data Sources

Patient information was collected through an online questionnaire (Appendix A), distributed to members of existing PASC associations in Spain. The questionnaire was designed and approved by members of the multiprofessional teaching unit of family and community care of the Córdoba and Guadalquivir health district, with established expertise in survey design and validation. The questionnaire underwent a consensus, apparent logic, and content validation process (face validity). The fieldwork (the period allowed for questionnaire completion) was conducted between March 2021 and January 2022.

### 2.5. Study Variables

The study variables were grouped into the following categories:Sociodemographic data: age, gender, population, area of residence (urban, 20,000 inhabitants or more vs. rural, less than 20,000 inhabitants);Work-related data: currently employed or on sick leave due to PASC;Clinical data: comorbidity, toxic habits (tobacco consumption and alcohol intake), duration of the long COVID19 (PASC) condition, persistent COVID symptoms and signs, incapacitating symptoms of PASC (fatigue, impaired ability to concentrate, brain fog, muscle pain, mental confusion, general discomfort, memory loss, joint pain, headache, difficulty breathing or dyspnea, difficulty sleeping, dizziness, chest pressure, back pain, chest pain, palpitations, dizziness, post-traumatic stress disorder, ocular discomfort, paresthesias, abdominal pain, dry eyes, diarrhea, stomachache, tinnitus, nausea, fever, cough, tremors, hoarseness, chills, high blood pressure, sweating, hypothermia, swelling, nasal congestion, difficulty swallowing, itching, loss of smell, low blood pressure, sore throat, hair loss, hives or eczema on the skin, loss of taste, vomiting, acrosyndrome, sputum or phlegm, loss of appetite, weight loss, facial erythema, seizures, and hemoptysis);Self-perceived health status (evaluated on a Likert scale from 1 to 10, with 1 being the worst perceived health status and 10 being the best perceived health status);Quality of life (evaluated on a Likert scale from 1 to 10, with 1 being the worst perceived quality of life and 10 being the best perceived quality of life).

### 2.6. Statistical Analysis

Various statistical methods were employed to analyze the involved variables. For qualitative variables, absolute and relative frequencies were analyzed. For quantitative variables, median and interquartile range were calculated.

In bivariate analysis, specific tests detailed in the corresponding result tables were employed. In brief, the χ^2^ (chi-square) test was used to compare proportions, or the Fisher exact test if any observed frequency was less than 5 cases. For comparing quantitative variables, the Wilcoxon rank sum test was preferred for its robustness compared to the Student’s *t*-test.

Multivariate analysis was conducted through binary logistic regression, using the “sick leave due to PASC” variable as the dependent variable. Selected independent variables included those with clinical relevance for researchers (age and gender) and those showing a *p*-value < 0.1 in bivariate analysis. Regression results are presented with odds ratios (OR) and their confidence intervals at a 95% confidence level, along with *p*-values obtained.

For the analysis, R (version “Bird Hippie” 4.1.2 (1 November 2021)) and RStudio (version “Elsbeth Geranium” (2022.12.0. + 353)) were used. Packages employed included car, crayon, dplyr, flextable, forcats, glue, ggplot2, gtsummary, haven, janitor, labelled, lubridate, nortest, pls, purrr, readx, stats, stringr, summarytools, tibble, and tidyr.

### 2.7. Ethical Considerations

The research project has the authorization of the managing directors of the Córdoba and Guadalquivir Health District and the Southern Córdoba Health Area of the Andalusian Health Service, as well as the approval of the Clinical Research Ethics Committee of the Reina Sofía Hospital in Córdoba (reference: 5033). Informed consent, ensuring voluntary participation of study patients, was obtained. Data treatment adhered to the provisions of the European Data Protection Regulation and Spanish Organic Law 3/2018 on the protection of personal data and guarantee of digital rights.

## 3. Results

The final sample size comprised 689 Spanish subjects, mostly women (83.2%). The median age was 46 years, higher in males (50 years, *p* = 0.005). A total of 10.01% (69) were smokers. Regarding pre-existing conditions prior to the onset of persistent COVID-19, noteworthy findings include: 19% (128) had been diagnosed with hypertension (HTN); 4.5% (31) with diabetes mellitus (DM); 27% (185) with hyperlipidemia; and 34% (235) with overweight or obesity. Concerning demographic characteristics, 80% (548) lived in urban areas (Table 1), mostly in the regions of Madrid and Andalusia.

Other chronic concomitant conditions are described in Table 2.

Consideration of variables related to COVID-19, diagnosis was primarily conducted through PCR (410). As many as 30% (205) experienced pneumonia following COVID-19, 23% (161) required hospitalization, while 3.6% (25) needed ICU admission (Table 3).

Participants reported a mean perceived disability due to COVID-19 of 7.0 (IQR 5, 8) and a QoL of 8.0 (IQR 7, 9), with no significant differences between men and women (Table 4).

When asked about persistent or recurrent symptomatology, the most frequent symptoms (%, n) were fatigue (89%, 616), brain fog (79%, 543), and muscle pain (77%, 528) (Table 5). Men reported shortness of breath (75% vs. 64%), chest tightness (59% vs. 49%), and chest pain (54% vs. 43%) more frequently than women.

The respondents pointed out that fatigue (86%, 594), lack of concentration/impaired attention (69%, 476), and brain fog (435, 63%) were the most disabling symptoms affecting their daily activities (Table 6). When comparing by gender, a higher percentage of dizziness was observed in males (52% vs. 41%; *p* = 0.04). No other statistically significant differences were found.

Table 7 displays the variables that, in the bivariate analysis, showed a statistically significant association with having been on sick leave due to PASC.

As a final point, a multivariate analysis was conducted to explore the association of various variables with “Sick leave due to persistent COVID-19” (Table 8). Patients who reported having suffered renal failure (OR: 4.04; CI95%: 1.42–11.4; *p*: 0.008) or tremors (OR: 2.0; CI95%: 1.06–3.69; *p*: 0.029) were more likely to have been on sick leave. On the other hand, those who reported a time since diagnosis of persistent COVID-19 ≥ 361 days (OR: 0.57; CI95%: 0.36–0.89; *p*: 0.013), and those with fatigue (OR: 0.56; CI95%: 0.33–0.99) were less likely to have been on sick leave.

## 4. Discussion

The objective of this study was to describe the sociodemographic and clinical profile of Spanish people suffering PASC and analyze its association with work-related incapacity.

As identified by other authors [7,8,9], the majority of the study population with PASC were women, with a median age of 46 years, differing by gender (50 years for men). Accordingly, most of the subjects studied were of working age. They predominantly were residents in urban areas, with a wide representation from the different regions (autonomous communities) of Spain, with a higher proportion from the two most populated regions, Madrid and Andalusia.

The findings of this multicenter study indicate that fatigue and lack of concentration/mental fog were the most debilitating symptoms among patients diagnosed with PASC. Furthermore, over half of the analyzed population (68%) reported being on medical leave due to PASC, with no gender-based differences. In terms of occupational impact, the literature suggests that a majority of Spanish patients with PASC (60%) had not returned to work between two and three months post-infection [18]. Given that estimates propose PASC could affect between 450,000 and 900,000 people in Spain [19], there is a significant number of patients who remain work-inactive due to this syndrome, posing a significant health challenge requiring both scientific and medical responses. Additionally, this bears social and economic implications in terms of social security benefits, with an estimated cost of around 1.8 billion euros by 2021 for COVID-19-related benefits alone [20].

From an occupational health standpoint, medical leave related to PASC should be considered as “prolonged leave”, exceeding not only the threshold of at least four weeks initially considered in the CDC’s PASC concept [21], but even the eight-week threshold proposed by the WHO [2]. Most of the participants in the survey said that they were or had been on sick leave, and the time since the COVID-19 diagnosis was longer than 361 days. It is important to note that, in Spain, when medical leave surpasses 365 days, it falls under the purview of the medical inspection of the National Institute of Social Security (INSS, in Spanish, Instituto Nacional de la Seguridad Social), potentially leading to a proposal for permanent disability to work. However, such a situation might not be ideal as it excludes the affected individual from active work life and social interactions in their work environment. In this context, being categorized under prolonged medical leave is not trivial; this leads primary care physicians to contemplate whether the patient still has limiting symptoms preventing an early return to work, and upon deciding on medical discharge, an occupational health check-up for the PASC patient is required after work reintegration. This medical assessment is conducted by the occupational health units’ healthcare personnel, which should result in adaptation measures, mandatory for the employer, to facilitate the PASC patient’s reintegration back into the work environment. For instance, this might involve a gradual increase in workload to facilitate adaptation. Certain studies show that 45.2% of PASC patients required a reduction in work hours, and 22.3% were not working due to the illness [13]. All of this translates into a loss of work capacity compared to their state prior to infection. This new functional capacity must be evaluated by the medical professionals of the risk prevention services, who have an established protocol [22] for a gradual return to an active work situation.

Permanent disability to work should be the last resort, as it results in early retirement for the affected individual. Therefore, it is important to adequately inform both workers and employers, as well as social agents, of the right of the PASC patient to both appropriate medical leave and an immediate occupational health examination upon work reintegration, along with adaptation measures to facilitate their return to work. These essential measures are critical for the care of PASC patients with a view to their recovery and socioeconomic reintegration. Such information about their occupational health rights should be systematically provided by primary care professionals and mutual insurance companies overseeing their medical leave process, with their work and coordination being essential in this context.

On another note, one might consider whether PASC is classified as an occupational contingency in Spain. The INSS refers to PASC or “prolonged COVID” as “a pathological entity that implies physical, medical, and cognitive/psychological sequelae after having experienced a COVID-19 infection with repercussions on various organs and systems” [23]. Emphasis is placed on “always derived from a previous diagnosis of COVID-19 disease”. The mention of “disease” would suggest its recognition as an “occupational disease”, since the causative virus is included among the biological agents contemplated in Spanish specific legislation [24]. However, the criteria for contingency of the INSS stipulate that COVID-19 disease has been recognized as requiring economic benefits for leave similar to workplace accidents [25]. This exceptional provision was designed to curb the spread of COVID-19. Nonetheless, since PASC is not a “relapse” of the same disease following acute infection by the virus, patients suffering from it must “be subject to general sick leave regulations” [26]. This implies that PASC would not be considered a workplace accident, except for the exceptional situation of healthcare or social healthcare personnel when the cause of the contagion is demonstrated to have occurred during their professional duties [25].

In relation to the clinical profile, the results are consistent with other authors [3,9,27] and show a heterogeneity of symptoms, with “fatigue” at 86% being the most persistent symptom from the virus infection, in line with the information published by WHO and CDC [2,21]. Fatigue, which some authors have associated with “Chronic Fatigue Syndrome”, is also the most persistent and limiting symptom, influencing the working capacity and daily life of affected individuals [28]. The higher percentage of “fatigue” found in our study (86%) might be due to respondents equating the symptom of “asthenia” (resting tiredness) with “fatigue” (reduced threshold for physical and/or intellectual exercise). The persistence of such fatigue is a relevant variable when dealing with work demanding functional capacity. However, as it is a subjective symptom, it is possible that it has influenced the fact that many of these patients have not been on sick leave due to this symptomatology, either because they have not requested it from their doctor or because the doctor has not considered it a sufficient reason to give them sick leave. Other symptoms that mostly persisted in our study align with WHO and CDC, including “lack of concentration/brain fog”, “muscle pain”, “shortness of breath”, “chest tightness”, “chest pain”, “cough”, “ageusia”, and “anosmia”. Differences by gender were observed for symptoms related to the chest (more frequently reported by men than women). Moreover, the majority of individuals with PASC surveyed reported experiencing anxiety (45%), overweight or obesity (34%), depression (27%), hyperlipidemia (27%), neurological disease (25%), and asthma (20%) as prevalent conditions. It is important to discern whether these conditions were pre-existing to the infection to rule out PASC.

In this context, the recent literature [19] highlights the need for a comprehensive differential diagnosis to rule out clinical manifestations falling into the categories of: (a) sequels from organ damage resulting from acute COVID-19 infection, (b) those arising from hospitalization, (c) those stemming from exacerbation of pre-existing chronic conditions, (d) those from the onset of autoimmune, metabolic, or psychiatric diseases triggered by COVID-19, and (e) effects of treatments administered during hospitalization of PASC patients. Boix and Merino [19] mention in their study that clinical manifestations that cannot be grouped into these categories and persist over time constitute PASC as a post-infectious syndrome in itself. This is a complex matter, not just due to the difficulty in interpreting symptomatology, but also because only a third of diagnoses usually include a clinical interview [19]. In fact, in our study, only 19% of PASC cases were diagnosed “clinically”, with most diagnosed through PCR testing. Additionally, these authors’ proposal contradicts the INSS’s definition of PASC, which encompasses both “sequelae” and “repercussions on organs and systems” [23]. The INSS, in essence, is the institution that in Spain administers benefits arising from medical leave and, potentially, from permanent disability to work due to PASC.

It is also worth considering the “persistence over time”, as the literature shows that the incidence of PASC decreases over time. For instance, a recent prospective cohort study conducted by Huang et al., involving over 1200 PASC patients, indicates symptom presence in 68% of patients at six weeks, decreasing to 49% at twelve weeks post-diagnosis [11]. This suggests a clinical follow-up at twelve weeks, during which basic health variables should be measured and controlled, such as blood pressure, kidney function, or neurological symptoms (asking about “dizziness” and “tremors”), aligning with results found in our study. This could also serve as an opportune moment for physicians to discuss with patients when they envision reintegration into their workplace, in coordination with healthcare personnel from risk prevention services.

Finally, regarding self-perceived health and quality of life, the study’s results show low self-rated health status, a deterioration in both health status and quality of life, and significant changes in the degree of disability upon relating it to the variable “medical leave due to PASC”. The predominant symptomatology (fatigue, brain fog), pathologies like pneumonia (experienced by almost a third of respondents), a significant relationship with renal insufficiency, or hospitalization (nearly a quarter) could explain this. These results are similar to those published in previous studies in Spanish PASC patients [29,30]. In line with the work-health binomial, we suggest, for the enhancement of quality of life, greater coordination not only between primary and specialized care for optimal attention to these patients [19], but also between the care provided by the national health system (responsible for healthcare), the benefits system (INSS, Workers’ Mutuals), and “the preventive occupational health system” (basic occupational health units from the risk prevention services).

In this regard, and with a focus on discharging and reintegrating the “particularly sensitive worker” PASC patient back into their workplace, we draw attention to the importance of evaluating the specific dysfunctional intensity of symptoms and functional work capacity of the PASC patient to contrast with the demands of their job, and implementing specific workplace programs for functional readaptation of these individuals according to their most debilitating symptoms. This should be guided by policies [31] and specific procedures [22,32] that incorporate tests and/or questionnaires regarding clinical presentation. We propose that personnel could be classified based on “sentinel symptoms” among non-sedentary (fatigue, muscle pain) and sedentary jobs (concentration deficits/brain fog), initiating specific programs of controlled and progressive physical exercise adapted to the individual (for non-sedentary roles), and cognitive skills (for sedentary roles). The objective would be to prevent disability, which has already been acknowledged in PASC patients by the American Disability Act [33]. Additionally, and as Godeau et al. mention, the use of scales such as the work ability index, or the work productivity and activity impairment, can help perform long-term follow-up and provide information about work capacity and workload [34].

Furthermore, there is significance in the treatment and care provided by primary care professionals (physicians and nurses) for pre-existing conditions indicated in the study’s results (HTN, DM, hyperlipidemia, overweight or obesity), all of which, apart from being risk factors for PASC [7,8,9], are interconnected through another syndrome, namely the metabolic syndrome. All these conditions are manageable with proper treatment and monitoring to avoid a poor prognosis for PASC.

The study has limitations, most of them inherent to survey-based descriptive research. As the questionnaire was massively distributed through the long COVID patient associations, it was not possible to determine the response rate. On the other hand, as in all studies carried out through surveys offered to the population, the lack of interest and motivation for the topic may lead to a selection bias. This may mean that patients who are more sensitized to the topic (e.g., those who perceive that they have been more affected by the disease) are more likely to participate. Also, as the questionnaire was self-reported by patients (patients’ clinical records were not reviewed as the questionnaires were anonymous), there may have been under-reporting of symptoms that had less impact on the patient. Similarly, some of the self-reported symptoms could be due to other concomitant conditions and be wrongly attributed to COVID-19. Another bias to be considered could be that of memory, although as it is a rather disabling condition, it is to be expected that this has been fairly minimized.

One of the main strengths of the study is that it constitutes multicenter research that analyzes in a pioneering way the perception of disability and its impact on the state of health of individuals suffering from a major health problem, often little understood by the scientific community and society in general, proposing practical actions in response to the explicit need to carry out “studies to understand the real impact, from the perspective of the PASC patient” [22].

## 5. Conclusions

The obtained results highlight that fatigue and lack of concentration/mental fog were the most incapacitating symptoms among patients diagnosed with post-acute sequelae of SARS-CoV-2 (PASC), who were predominantly women, residing in urban areas, and in their fifth decade of life.

This study confirms the heterogeneity of the clinical profile as demonstrated by previous research. Regarding the work incapacity of the studied PASC patients, variables of comorbidity (renal insufficiency), and symptomatology (tremor) showed a positive association with medical leave in multivariate analysis.

In addition to monitoring basic parameters like renal function or variables comprising the metabolic syndrome, it is suggested that primary care healthcare professionals should pay special attention to the timing of issuing medical leave for these patients. This involves relating the specific dysfunctional intensity of symptoms to the corresponding work functionality of the PASC patient. Coordination with the occupational health prevention system is also recommended for a gradual and tailored reintegration of PASC patients (a right they have as workers with prolonged medical leave) to prevent early permanent disability.

## Figures and Tables

**Table 1 healthcare-11-02632-t001:** Age and most relevant cardiovascular risk factors, differentiated by gender.

Variable	Overall, N = 689 ^1^	Men, N = 116 ^1^	Women, N = 573 ^1^	*p*-Value
Age (yr)	46 (40, 52)	50 (42, 54)	46 (40, 51)	0.005 ^2^
Geographical area of residence				0.3 ^3^
Urban	548 (80%)	96 (83%)	452 (79%)	
Rural	141 (20%)	20 (17%)	121 (21%)	
Region (Autonomous Community) of residence				0.2 ^4^
Madrid	177 (26%)	29 (25%)	148 (26%)	
Andalusia	129 (19%)	25 (22%)	104 (18%)	
Aragon	84 (12%)	13 (11%)	71 (12%)	
Catalonia	73 (11%)	6 (5.2%)	67 (12%)	
Valencian	42 (6.1%)	5 (4.3%)	37 (6.5%)	
Other	184 (27%)	38 (33%)	146 (25%)	
Unknown				
Smoker				0.14 ^4^
Never smoked	364 (53%)	72 (62%)	292 (51%)	
Former smoker	256 (37%)	37 (32%)	219 (38%)	
Occasional	23 (3.3%)	3 (2.6%)	20 (3.5%)	
Daily	46 (6.7%)	4 (3.4%)	42 (7.3%)	
High Blood Pressure	128 (19%)	23 (20%)	105 (18%)	0.7 ^3^
Diabetes Mellitus	31 (4.5%)	6 (5.2%)	25 (4.4%)	0.7 ^3^
Hyperlipidemia	185 (27%)	37 (32%)	148 (26%)	0.2 ^3^
Overweight or Obesity	235 (34%)	46 (40%)	189 (33%)	0.2 ^3^

^1^ Median (IQR); n (%). ^2^ Wilcoxon rank sum test. ^3^ Pearson’s Chi-squared test. ^4^ Fisher’s exact test.

**Table 2 healthcare-11-02632-t002:** Reported concomitant conditions, other than COVID-19.

Variable	Overall, N = 689 ^1^	Men, N = 116 ^1^	Women, N = 573 ^1^	*p*-Value
COPD	25 (3.6%)	1 (0.9%)	24 (4.2%)	0.10 ^2^
Asthma	136 (20%)	20 (17%)	116 (20%)	0.5 ^3^
Immunosuppression	110 (16%)	16 (14%)	94 (16%)	0.5 ^3^
Cancer	8 (1.2%)	2 (1.7%)	6 (1.0%)	0.6 ^2^
Respiratory insufficiency	130 (19%)	21 (18%)	109 (19%)	0.8 ^3^
Renal insufficiency	24 (3.5%)	5 (4.3%)	19 (3.3%)	0.6 ^2^
Heart failure	39 (5.7%)	7 (6.0%)	32 (5.6%)	0.8 ^3^
Hepatic insufficiency	16 (2.3%)	3 (2.6%)	13 (2.3%)	0.7 ^2^
Depression	187 (27%)	30 (26%)	157 (27%)	0.7 ^3^
Anxiety	312 (45%)	54 (47%)	258 (45%)	0.8 ^3^
Mental illness ^4^	33 (4.8%)	4 (3.4%)	29 (5.1%)	0.5 ^3^
Autoimmune disease ^5^	89 (13%)	14 (12%)	75 (13%)	0.8 ^3^
Vascular disease ^6^	35 (5.1%)	4 (3.4%)	31 (5.4%)	0.4 ^3^
Heart disease ^7^	62 (9.0%)	5 (4.3%)	57 (9.9%)	0.053 ^3^
Endocrine disorder ^8^	124 (18%)	28 (24%)	96 (17%)	0.059 ^3^
Neurological disease	174 (25%)	30 (26%)	144 (25%)	0.9 ^3^

^1^ n (%). ^2^ Fisher’s exact test. ^3^ Pearson’s Chi-squared test. ^4^ Mental illness (psychosis, neurosis, Alzheimer’s, etc.). ^5^ Autoimmune disease (ulcerative colitis, etc.). ^6^ Vascular disease (stroke, cerebrovascular accident, arteriopathy, etc.). ^7^ Heart disease (atrial fibrillation, valvulopathy, myocardial infarction, angina, left ventricular hypertrophy, etc.). ^8^ Endocrine disorder (hypothyroidism, etc.).

**Table 3 healthcare-11-02632-t003:** Other variables related to COVID-19 diagnosis and outcome.

Variable	Overall, N = 689 ^1^	Men, N = 116 ^1^	Women, N = 573 ^1^	*p*-Value
Diagnosed via PCR	410 (60%)	61 (53%)	349 (61%)	0.10 ^2^
Diagnosed clinically	128 (19%)	21 (18%)	107 (19%)	0.9 ^2^
Diagnosed using rapid antigen test	117 (17%)	27 (23%)	90 (16%)	0.048 ^2^
Diagnosed through serological test	85 (12%)	18 (16%)	67 (12%)	0.3 ^2^
Pneumonia following COVID-19 diagnosis	205 (30%)	35 (30%)	170 (30%)	>0.9 ^2^
Hospitalized due to COVID-19	161 (23%)	28 (24%)	133 (23%)	0.8 ^2^
Admitted to ICU due to COVID-19	25 (3.6%)	3 (2.6%)	22 (3.8%)	0.8 ^3^
Time since post-acute sequelae SARS-CoV-2 infection (PASC)				>0.9 ^2^
<361 dy	216 (31%)	36 (31%)	180 (31%)	
≥361 dy	473 (69%)	80 (69%)	393 (69%)	
Work leave due to PASC	467 (78%)	75 (72%)	392 (80%)	0.089 ^2^
Unknown	93	12	81	
Vaccinated for SARS-CoV-2	602 (87%)	101 (87%)	501 (87%)	>0.9 ^2^

^1^ n (%). ^2^ Pearson’s Chi-squared test. ^3^ Fisher’s exact test.

**Table 4 healthcare-11-02632-t004:** Health perception and quality of life assessment related to COVID-19.

Variable	Overall, N = 689 ^1^	Men, N = 116 ^1^	Women, N = 573 ^1^	*p*-Value
Self-perceived current general health status (0 to 10) *	4.00 (3.00, 6.00)	4.00 (3.00, 5.00)	4.00 (3.00, 6.00)	0.3 ^2^
Degree of change in self-perceived health status compared to before contracting COVID-19 (0 to 10)	8.00 (7.00, 9.00)	8.00 (7.00, 9.00)	8.00 (7.00, 9.00)	0.8 ^2^
Degree of self-perceived disability due to persistent COVID-19 (0 to 10)	7.00 (5.00, 8.00)	8.00 (5.00, 8.00)	7.00 (5.00, 8.00)	0.8 ^2^
Degree of self-perceived impact on quality of life (0 to 10)	8.00 (7.00, 9.00)	8.00 (7.00, 9.00)	8.00 (7.00, 9.00)	0.2 ^2^

^1^ Median (IQR). ^2^ Wilcoxon rank sum test. * Higher values indicate worse perception.

**Table 5 healthcare-11-02632-t005:** Persistent or recurrent symptoms since diagnosis of COVID-19 (reported by >50% of subjects or with a statistical difference by gender).

Variable	Overall, N = 689 ^1^	Men, N = 116 ^1^	Women, N = 573 ^1^	*p*-Value
Fatigue	616 (89%)	108 (93%)	508 (89%)	0.2 ^2^
Brain fog	543 (79%)	94 (81%)	449 (78%)	0.5 ^2^
Muscle pain	528 (77%)	93 (80%)	435 (76%)	0.3 ^2^
Joint pain	499 (72%)	89 (77%)	410 (72%)	0.3 ^2^
General malaise	487 (71%)	88 (76%)	399 (70%)	0.2 ^2^
Memory impairment	484 (70%)	86 (74%)	398 (69%)	0.3 ^2^
Sleeping difficulties	472 (69%)	85 (73%)	387 (68%)	0.2 ^2^
Mental confusion	466 (68%)	87 (75%)	379 (66%)	0.063 ^2^
Sensation of shortness of breath, dyspnea	453 (66%)	87 (75%)	366 (64%)	0.021 ^2^
Concentration deficit	431 (63%)	74 (64%)	357 (62%)	0.8 ^2^
Back pain	372 (54%)	68 (59%)	304 (53%)	0.3 ^2^
Dizziness	367 (53%)	70 (60%)	297 (52%)	0.094 ^2^
Difficulty breathing or breathlessness	365 (53%)	68 (59%)	297 (52%)	0.2 ^2^
Chest tightness sensation	349 (51%)	69 (59%)	280 (49%)	0.037 ^2^
Chest pain	312 (45%)	63 (54%)	249 (43%)	0.032 ^2^

^1^ n (%). ^2^ Pearson’s Chi-squared test.

**Table 6 healthcare-11-02632-t006:** Most disabling symptoms affecting their daily activities (reported by >50% of subjects or with a statistical difference by gender).

Variable	Overall, N = 689 ^1^	Men, N = 116 ^1^	Women, N = 573 ^1^	*p*-Value
Fatigue	594 (86%)	101 (87%)	493 (86%)	0.8 ^2^
Concentration deficit	476 (69%)	85 (73%)	391 (68%)	0.3 ^2^
Brain fog	435 (63%)	79 (68%)	356 (62%)	0.2 ^2^
Muscle pain	426 (62%)	72 (62%)	354 (62%)	>0.9 ^2^
Mental confusion	417 (61%)	78 (67%)	339 (59%)	0.10 ^2^
General malaise	407 (59%)	71 (61%)	336 (59%)	0.6 ^2^
Memory impairment	399 (58%)	68 (59%)	331 (58%)	0.9 ^2^
Joint pain	378 (55%)	67 (58%)	311 (54%)	0.6 ^3^
Sensation of shortness of breath, dyspnea	378 (55%)	73 (63%)	305 (53%)	0.055 ^2^
Headache	360 (52%)	64 (55%)	296 (52%)	0.5 ^2^
Dizziness	297 (43%)	60 (52%)	237 (41%)	0.040 ^2^

^1^ n (%). ^2^ Pearson’s Chi-squared test. ^3^ Fisher’s exact test.

**Table 7 healthcare-11-02632-t007:** Bivariate analysis of possible variables associated with the occurrence of sick leave (with a *p*-value ≤ 0.10).

Variable	Overall *, N = 596 ^1^	No Sick Leave, N = 129 ^1^	Some Sick Leave, N = 467 ^1^	*p*-Value
Age (yr)	46 ^1^ (40, 52)	44 (38, 51)	47 (41, 52)	0.024 ^2^
Gender				0.089 ^3^
Men	104 (17%)	29 (22%)	75 (16%)	
Women	492 (83%)	100 (78%)	392 (84%)	
Immunosuppression	92 (15%)	14 (11%)	78 (17%)	0.10 ^3^
Renal insufficiency	18 (3.0%)	8 (6.2%)	10 (2.1%)	0.035 ^4^
Heart disease (atrial fibrillation, valvulopathy, myocardial infarction, angina, left ventricular hypertrophy, …)	54 (9.1%)	7 (5.4%)	47 (10%)	0.10 ^3^
Time since post-acute sequelae SARS-CoV-2 infection (PASC)				0.001 ^3^
<361 dy	181 (30%)	54 (42%)	127 (27%)	
≥361 dy	415 (70%)	75 (58%)	340 (73%)	
Fatigue	513 (86%)	102 (79%)	411 (88%)	0.009 ^3^
Mental confusion	351 (59%)	66 (51%)	285 (61%)	0.044 ^3^
Post-traumatic stress	141 (24%)	21 (16%)	120 (26%)	0.026 ^3^
Tremors	71 (12%)	21 (16%)	50 (11%)	0.084 ^3^

^1^ Median (IQR); n (%). ^2^ Wilcoxon rank sum test. ^3^ Pearson’s Chi-squared test. ^4^ Fisher’s exact test. * Respondents who were active workers during the course of the disease (not unemployed or retired).

**Table 8 healthcare-11-02632-t008:** Multivariate analysis using logistic regression for the variable ‘Sick leave due to persistent COVID-19’.

Variable	OR ^1^	95% CI ^2^	*p*-Value
Age (per year of increment)	0.98	0.96, 1.01	0.13
Being a woman	0.64	0.39, 1.08	0.090
Having immunosuppression	0.64	0.32, 1.19	0.2
Having renal insufficiency	4.04	1.42, 11.4	0.008
Having heart disease	0.56	0.22, 1.25	0.2
Time since post-acute sequelae SARS-CoV-2 infection (PASC) ≥ 361 days	0.57	0.36, 0.89	0.013
Having fatigue	0.56	0.33, 0.99	0.041
Having mental confusion	0.73	0.47, 1.13	0.2
Having post-traumatic stress	0.59	0.33, 1.01	0.063
Having tremors	2.00	1.06, 3.69	0.029

^1^ OR = Odds Ratio. ^2^ CI = Confidence Interval.

## Data Availability

The datasets used and analyzed during the current study are available from the corresponding authors on reasonable request.

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
