# Peer review of "Sociodemographic and Clinical Profile of Long COVID-19 Patients, and Its Correlation with Medical Leave: A Comprehensive Descriptive and Multicenter Study"

_healthcare, 2023, doi:10.3390/healthcare11192632_

Round 1

Reviewer 1 Report

Dear Authors,

thank you for your interesting manuscript entitled " Sociodemographic and clinical profile of Long COVID-19 patients, and its correlation with medical leave: A comprehensive descriptive and multicenter study "

I have some comments that I hope that it may help

1- you did correlate the long-COVID with many diseases,  and since the mechanism of long-COVID is not established yet, I wish that you can subgroup the patients regarding to the age, familial history of chronic diseases and the duration between time of infection and long-COVID manifestations.

2- since the title mentioned the social factor in the study, I am looking forward of parameters correlated to the social status (like income, nutrition, ...etc).

More clarifying to the results will clarify the significance of data.

Reviewer 2 Report

The manuscript, titled “Sociodemographic and clinical profile of Long COVID-19 patients, and its link with medical leave: A comprehensive descriptive and multicenter study,” seeks to assess the impact of post-COVID-19 sequelae on quality of life, perceived health status, and the workplace. The study is well-done, with valuable methods employed, and the introduction looks well-argued. I only have a few comments and considerations to provide.

2. Materials and Methods

2.1 Study Design: Given what you indicated, it would be useful to know which hospitals participated in the multicenter survey.

2.4 Data Sources: I attempted, but failed, to access the questionnaire. Could you please include it as a Supplementary File? It would be beneficial to researchers who wish to conduct another study based on your survey and strengthen the evidence you obtained.

2.5 Study Variables: I am interested if you have any geographical data that may be examined to confirm any differences. It would also be interesting to know if you have data pertaining to the period of infection, so that you may estimate the variant with which individuals are most likely infected. The variant evaluation (presumed, not certain, unless you have specific data on all participants recruited for the survey (I read, in section 2.2, about “(c) Laboratory-diagnosed acute COVID-19 infection”) might be extremely important information, especially if there are changes in symptoms and so on.

I would also like to know why you did not employ quality-of-life questionnaires that have already been validated. There is, in particular, a survey – the WHO-QoL-BREF, available at https://www.who.int/tools/whoqol/whoqol-bref – that has been translated and validated into almost all languages. It could have been a reason and a chance to compare the findings to other studies.

3. Results

Table 8: I am perplexed by the Fatigue data as well as the pattern of other comorbidities, except for Tremors. In the meantime, I would like you to improve the table. Try to retain only the reference that is used for each variable (for example, “per unit increment” for age; “presence of...” for comorbidities and symptoms if the reference is the presence of and I understood correctly). Also, I would like to understand your perspective because it appears that the presence of multiple comorbidities and symptoms reduces the likelihood of leaving a job.

Reviewer 3 Report

Dear authors, congratulations on the study! In general, your manuscript has quality for a publication. Although your data do not bring a large context of "new" information, I believe that the dissemination of information on the COVID topic continues to be important.

Suggestions for the study:

1. The end of Table 2 is a bit polluted. This makes reading difficult. I suggest improving the presentation of data by reducing the number of information. Authors can put in the legend (below the Table) the information between relatives about: Mental illness, Autoimmune disease, etc;

2. Considering the large amount of information available on the impact of the corona virus on populations, I suggest that a strong point of the present study is to bring to light information on a specific population: highlighting Spain and the residences of origin of the population that responded to the survey.

Round 2

Reviewer 1 Report

Dear Authors,

Thank you for updating the manuscript.

Good Luck